# A Series of Green Oxovanadium(IV) Precatalysts with O, N and S Donor Ligands in a Sustainable Olefins Oligomerization Process

**DOI:** 10.3390/molecules27228038

**Published:** 2022-11-19

**Authors:** Mariusz Urbaniak, Kacper Pobłocki, Paweł Kowalczyk, Karol Kramkowski, Joanna Drzeżdżon, Barbara Gawdzik, Patrycja Świtała, Maja Miler, Daria Heleniak, Przemysław Rybiński, Dagmara Jacewicz

**Affiliations:** 1Institute of Chemistry, Jan Kochanowski University, Uniwersytecka 7, 25-406 Kielce, Poland; 2Department of Environmental Technology, Faculty of Chemistry, University of Gdansk, Wita Stwosza 63, 80-308 Gdansk, Poland; 3Department of Animal Nutrition, The Kielanowski Institute of Animal Physiology and Nutrition, Polish Academy of Sciences, Instytucka 3, 05-110 Jabłonna, Poland; 4Department of Physical Chemistry, Medical University of Bialystok, Kilińskiego, 15-089 Białystok, Poland

**Keywords:** oligomerization, oxovanadium(IV) complexes, olefins, catalysis, green chemistry

## Abstract

Designing catalyst systems based on transition metal ions and activators using the principles of green chemistry is a fundamental research goal of scientists due to the reduction of poisonous solvents, metal salts and organic ligands released into the environment. Urgent measures to reduce climate change are in line with the goals of sustainable development and the new restrictive laws ordained by the European Union. In this report, we attempted to use known oxovanadium(IV) green complex compounds with O, N and S donor ligands, i.e., [VO(TDA)phen] • 1.5 H_2_O (TDA = thiodiacetate), (phen = 1,10-phenanthroline), oxovanadium(IV) microclusters with 2-phenylpyridine (oxovanadium(IV) cage), [VOO(dipic)(2-phepyH)] • H_2_O (dipic = pyridine-2,6-dicarboxylate anion), (2-phepyH = 2-phenylpyridine), [VO(dipic)(dmbipy)] • 2H_2_O (dmbipy = 4,4′-dimethoxy-2,2′-dipyridyl) and [VO(ODA)(bipy)] • 2 H_2_O (ODA = oxydiacetate), (bipy = 2,2′-bipyridine), as precatalysts in oligomerization reactions of 3-buten-2-ol, 2-propen-1-ol, 2-chloro-2-propen-1-ol and 2,3-dibromo-2-propen-1-ol. The precatalysts, in most cases, turned out to be highly active because the catalytic activity exceeded 1000 g mmol^−1^·h^−1^. In addition, the oligomers were characterized by Fourier-transform infrared spectroscopy (FTIR), matrix-assisted laser desorption/ionization (MALDI-TOF-MS), thermogravimetric analysis (TGA) and differential scanning calorimetry (DSC) techniques.

## 1. Introduction

The 20th century was the beginning of the era of polymers, which to a significant and increasing extent, have replaced other materials [1,2,3,4,5]. Polyolefins have the largest share in the global production of polymeric materials, including mainly polyethylenes and polypropylenes. Olefin polymerization is one of the most important commercial processes.

The process of olefin transformation was discovered in the 1950s in the petrochemical industry in the United States during research on olefin polymerization in the presence of Ziegler–Natta catalysts [6,7,8], which include TiCl_4_ and Et_2_AlCl. The main breakthrough, however, turned out to be the discovery of the so-called well-defined molybdenum catalysts in 1987 by Schrock [9,10] and then ruthenium complex compounds in 1992 by Grubbs [11,12,13]. These compounds have revolutionized chemical synthesis; in particular, organic synthesis and the synthesis of precursors of new materials. On the other hand, among the modified complex compounds, ruthenium catalysts of the second-generation Hoveyda–Grubbs type are very popular [14]. Second-generation Hoveyda–Grubbs type catalysts are very stable and can be separated from the reaction mixture by column chromatography using non-dried and non-oxygen-free solvents. The composition of the catalytic system, mainly the type of transition metal, the type of activator and, to a lesser extent, the ligand structure, have a significant impact on the activity of the catalytic system and the properties of the obtained oligomer or polymer [15,16,17].

In recent years, there has been great progress in the study of the mechanisms of the catalytic processes, as well as the increase in the efficiency and selectivity of the studied reactions. This progress is related mainly to the possibility of modifying the coordination sphere of precatalysts or catalysts through the selection of appropriate ligands. Thanks to such modifications, it is possible to obtain compounds with appropriate steric hindrance and electron density, i.e., catalysts with strictly defined and designed properties. From year to year, there is an increasing demand for environmentally friendly polymers, which should be characterized by favorable performance properties. As a result, they will contribute to the development of research on the development of new, effective organometallic catalysts for olefin polymerization. Low-pressure olefin polymerization with the use of organometallic catalysts and characterization of the properties of the obtained products are the subject of continuous research work, which significantly propels the development of existing polyolefin technologies and favors the continuous expansion of the directions of their application. Taking into account the above considerations, it can be concluded that new generation catalysts are currently being designed, which create unimaginable possibilities for industrial applications, in particular in the field of highly processed chemical compounds under the conditions of “green chemistry” [18,19,20,21]. The search for new, effective and environmentally friendly chemical technologies encourages greater use of biotechnological and organocatalytic methods [22,23,24,25].

Taking into account the above considerations, in this article, we would like to present the relationship between the structure of green (the synthesis of these complexes is carried out in the water, a green solvent and with vanadyl acetylacetonate as a metal salt) oxovanadium(IV) precatalysts with O, N and S donor ligands in the olefins oligomerization process and their catalytic properties. Catalytic systems based on green precatalysts, e.g., [VO(TDA)phen] • 1.5 H_2_O (TDA = thiodiacetate), (phen = 1,10-phenanthroline), oxovanadium(IV) microclusters with 2-phenylpyridine (oxovanadium(IV) cage), [VOO(dipic)(2-phepyH)] • H_2_O (dipic = pyridine-2,6-dicarboxylate anion), (2-phepyH = 2-phenylpyridine), [VO(dipic)(dmbipy)] • 2H_2_O (dmbipy = 4,4′-dimethoxy-2,2′-dipyridyl), [VO(ODA)(bipy)] • 2 H_2_O (ODA = oxydiacetate), (bipy = 2,2′-bipyridine) and ethylaluminium dichloride (AlEtCl_2_) as an activator, were tested in the oligomerization reaction of 2-propen-1-ol (allyl alcohol), 2-chloro-2-propen-1-ol, 3-buten-2-ol and 2,3-dibromo-2-propen-1-ol. This catalytic system is not reactive in the oligomerization reaction of 2-chloro-2-propen-1-ol. The results show that, in most cases, these are highly active precatalysts because their catalytic activity is above 1000 g mmol^−1^·h^−1^.

## 2. Results and Discussion

### 2.1. Oxovanadium(IV) Precatalysts 

The oxovanadium(IV) complex compounds (Figure 1) were synthesized according to the procedures described in the literature: ref. [26,27] for [VO(TDA)phen] • 1.5 H_2_O (Compound 1), ref. [28] oxovanadium(IV) microclusters with 2-phenylpyridine (oxovanaium(IV) cage) (Compound 2), ref. [29] for [VOO(dipic)(2-phepyH)] • H_2_O (Compound 3) and [VO(dipic)(dmbipy)] • 2H_2_O (Compound 4) and refs. [30,31] for [VO(ODA)(bipy)] • 2 H_2_O (Compound 5). All 5 compounds were used as precatalysts for olefin oligomerization reactions: 3-buten-2-ol, 2-chloro-2-propen-1-ol, 2-propen-1-ol (allyl alcohol) and 2,3-dibromo-2-propen-1-ol with ethylaluminium dichloride (AlEtCl_2_) as an activator. 

### 2.2. Analysis of FTIR Spectra of Olefin Oligomers 

The products of oligomerization were characterized by FTIR spectroscopy. The obtained results are presented in Table 1, Table 2 and Table 3. FTIR studies confirmed the structure of olefin oligomerization products. We also noticed that numerous hydrogen bonds were formed in the oligomer due to the lowering of the vibration frequency and the broadening of the band in the range of about 3000 cm^−1^ [32,33,34]. Additionally, characteristic bands for the V=O stretching frequency at approximately 980 cm^−1^ and the V-N stretching vibration at approximately 450 cm^−1^ were not observed [34,35]. Therefore, we can conclude that the olefin oligomers were well purified from oxovanadium(IV) complex compounds by the solution of hydrochloric acid and methanol. All FTIR spectra of the oligomerization products are available in the Appendix A.

### 2.3. MALDI-TOF-MS Spectra Analysis of the Oligomers 

Using the MALDI-TOF-MS method, we characterized certain peaks, thus allowing the identification of the number of units present in the oligomer chains using oxovanadium(IV) complex compounds as precatalysts and the degree of oligomerization (Table 4, Table 5 and Table 6) [36]. The matrix used for the measurements was 2,3-Dihydroxybenzoic acid (DHB). All MALDI-TOF-MS spectra of olefins oligomers are available in the Appendix A.

#### 2.3.1. Reaction of Oligomerization Catalyzed by [VO(TDA)phen] • 1.5 H_2_O

The 3-buten-2-ol oligomer consisted of a mixture of 5, 9, 11 and 14 mers. The value of the pseudo-molecular peak was 333 *m*/*z*. Most units were formed during the oligomerization of allyl alcohol. The mixture consisted of long strings of 7, 11, 14, 17 and 20 mers. In turn, the oligomer of 2,3-dibormo-2-propen-1-ol catalyzed by [VO(TDA)phen] • 1.5 H_2_O contained 2–4 units.

#### 2.3.2. Reaction of Oligomerization Catalyzed by [VOO(dipic)(2-phepyH)] • H_2_O

The oligomer of 3-buten-2-ol catalyzed by [VOO(dipic)(2-phepyH)] • H_2_O consisted of 3 and 6 mers. In the spectrum, one can distinguish the peak from the DHB matrix at 355 *m*/*z,* which corresponds to [2DHB + 2Na]. The MALDI-TOF-MS spectrum of the allyl alcohol oligomer shows a pseudo-molecular ion with a value of 411 *m*/*z*, which corresponds to a value of 7 mers of 2-propen-1-ol. Additionally, there was a mixture of 11, 14 and 17 mers in the oligomer. The shortest mer chains arose during the oligomerization reaction of 2,3-dibromo-2-propen-1-ol since only a mixture of 2 to 4 mers were specified.

#### 2.3.3. Reaction of Oligomerization Catalyzed by [VO(dipic)(dmbipy)] • 2 H_2_O

The oligomerization reaction of 3-buten-2-ol catalyzed by [VO(dipic)(dmbipy)] • 2 H_2_O produced mixtures of 3, 5 and 6 mers. In the oligomerization of allyl alcohol, the greatest number of units was formed because the mixture contained 2, 4, 6, 7, 8, 9 and 12 mers. In the MALDI-TOF-MS spectrum of the 2,3-dibormo-2-propen-1-ol oligomer, a template peak of 333 *m*/*z* [DHB + Mg] can be distinguished. Additionally, the oligomer contained 2–4 mers in the mixture.

#### 2.3.4. Reaction of Oligomerization Catalyzed by [VO(ODA)bipy] • 2 H_2_O

Of all the oligomerization reactions of 3-buten-2-ol, the longest mer chains were formed using [VO(ODA)bipy] • 2 H_2_O as a precatalyst. The pseudo-molecular ion could be specified at 411 *m*/*z,* corresponding to [5M], and the matrix at 333 *m*/*z* [DHB + Mg]. The 3-buten-2-ol oligomer consisted of a mixture of 5, 6, 9, 12 and 16 mers. In turn, during the oligomerization reaction of 2-propen-1-ol, a mixture of mers consisting of 7, 9, 10, 11, 14, 17 and 20 units was determined. The pseudo-molecular ion [7M] had a value of 411 *m*/*z*. The 2,3-dibromo-2-propen-1-ol oligomer consisted of 2 to 5 mers. The matrix peak value was 333 *m*/*z* and corresponded to [DHB + Mg].

### 2.4. Thermogravimetric Analysis (TGA) of the Oligomers 

#### 2.4.1. Reactions of Oligomerization Catalyzed by [VO(TDA)phen] • 1.5 H_2_O

A sample of the 3-buten-2-ol oligomer catalyzed by [VO(TDA)phen] • 1.5 H_2_O underwent two-step thermal decomposition [37]. The first was in the range of ∆T = 30–145 °C, while the second was in the range of ∆T = 145–240 °C. The rate of the second stage of thermolysis was 13.5%/min, and the decomposition residue was 27%. The residue after decomposition was incinerated in the range of ∆T = 240–300 °C. The residue after combustion was 16.6%, while the residue at T = 600 °C was 6.2%. 

The allyl alcohol oligomer catalyzed by [VO(TDA)phen] • 1.5 H_2_O underwent two-step chemical decomposition caused by heat. The first step of thermal decomposition took place in the range ∆T = 30–150 °C, while the second one was in the range ∆T = 150–245 °C. The rate of the second stage of thermal decomposition was 12.7%/min, and the decomposition residue was 26.5%. The residue after decomposition was incinerated in the range ∆T = 245–320 °C. The residue after combustion was 13.47%, while the residue at T = 600 °C was 6.7%. 

The 2,3-dibromo-2-propen-1-ol oligomer sample underwent single-step thermal decomposition in the range ∆T = 200–270 °C. The rate of thermal decomposition was 40.5 %/min, while the residue after decomposition was 21.95% at T = 600 °C.

#### 2.4.2. Reactions of Oligomerization Catalyzed by [VOO(dipic)(2-phepyH)] • H_2_O

The oligomer of 3-buten-2-ol catalyzed by [VOO(dipic)(2-phepyH)] • H_2_O underwent thermal decomposition in the temperature range ∆T = 150–260 °C. The maximum decomposition rate was 12.5%/min, while the residue after thermolysis was 33.9%. The residue after thermal decomposition was burned in the temperature range ∆T = 260–300 °C. The residue after combustion was 19.4%, while the residue at T = 600 °C was 9.1%. 

A sample of allyl alcohol oligomer underwent two-stage thermal decomposition. The first stage was registered in the temperature range ∆T = 30–140 °C, and the second in the range ∆T = 140–250 °C. The rate of the second stage of thermolysis was 12.5%/min, while the decomposition residue was 30.6%. The residue was incinerated in the range ∆T = 250–320 °C. The residue after combustion was 18.0%, while at T = 600 °C, it was 3.8%. 

The oligomer of the 2,3-dibromo-2-propen-1-ol underwent single-stage thermal decomposition in the range ∆T = 200–280 °C. The rate of chemical decomposition caused by heat was 32.90%/min, while the residue after decomposition was 18.1%. 

#### 2.4.3. Reactions of Oligomerization Catalyzed by [VO(dipic)(dmbipy)] • 2 H_2_O

The 3-buten-2-ol sample underwent single-step thermal decomposition which took place in the range ∆T = 130–265 °C. The rate of the first stage of thermal decomposition was 9.6%/min, and the decomposition residue was 34.5%. The residue after decomposition was incinerated in the range ∆T = 260–295 °C.

The allyl alcohol oligomer underwent two-stage thermal decomposition. The first took place in the range ∆T = 30–145 °C, while the second one was in the range ∆T = 152–265 °C. The rate of the second stage of thermal decomposition was 7.8%/min, and the decomposition residue was 38.4%. The residue after combustion at T = 600 °C was 32%.

The 2,3-dibromo-2-propen-1-ol oligomer underwent two-step thermal decomposition. The first was in the range of ∆T = 150–265 °C, while the second was in the range of ∆T = 340–423 °C. The rate of the second stage of thermal decomposition was 10.4%/min. The decomposition residue was burned in the ranges ∆T = 460–485 °C and at ∆T = 560–580 °C. The residue after combustion was 6.2%.

#### 2.4.4. Reactions of Oligomerization Catalyzed by [VO(ODA)bipy] • 2 H_2_O

The oligomer of 3-buten-2-ol underwent the main stage of thermal decomposition in the temperature range ∆T = 150–250 °C, and the thermal decomposition rate was 16.8%/min. The residue after thermolysis was burned in the range ∆T = 250–300 °C. The residue after combustion was 23.3%, while the residue at T = 600 °C was 13.9%.

The sample of allyl alcohol oligomer catalyzed by [VO(ODA)bipy] • 2 H_2_O underwent the initial stage of thermolysis in the temperature range ∆T = 50–150 °C. The essential stage of thermal decomposition took place in the range ∆T = 150–250 °C. The thermal decomposition rate was 15.7%/min. Combustion of the residue after chemical decomposition caused by heat took place in the temperature range ∆T = 250–305 °C. The residue after combustion was 22.3%, while the residue at T = 600 °C was 13.1%.

The oligomer of 2,3-dibromo-2-propen-1-ol oligomer underwent a single-stage, endothermic decomposition process in the range of ∆T = 180–280 °C. The rate of thermolysis was 25.76%/min, the residue after thermal decomposition was 26.7% and the residue at T = 600 °C was 12.3%.

### 2.5. Differential Scanning Calorimetry (DSC) Analysis of the Oligomers 

#### 2.5.1. Reactions of Oligomerization Catalyzed by [VO(TDA)phen] • 1.5 H_2_O

The glass transition temperature values for the 3-buten-2-ol oligomer ranged from ∆T = −132.5 to −126.1 °C (∆Cp = 0.03 J g^−1^ K^−1^); this step was followed by the oligomer transition from a glass to elastic state. There are three exothermic processes during heating. Two of them occur at T = 118.6 °C and T = 165.7 °C. The third, with the largest peak area (377 J g^−1^), occurs at ∆T = 211–237 °C and indicates the degradation of the residue after combustion. This is probably a symptom of the deoligomerization reaction of the 3-buten-2-ol oligomer [38,39].

The degradation of the allyl alcohol oligomer obtained using [VO(TDA)phen] • 1.5 H_2_O as a precatalyst took place in two stages, the at a temperature of 176.3 °C. The second stage took place at ∆T = 215–229 °C and indicated the degradation or deoligomerization of the allyl alcohol oligomer. No glass-to-elastic transition of the oligomer was observed in the DSC spectrum [38].

In the differential scanning calorimetry spectrum of 2,3-dibromo-2-propen-1-ol oligomer, a transition from an elastic to glassy state was noticed at ∆T = −116 to −73 °C (∆Cp = 0.07 J g^−1^ K^−1^). We observed two exothermic peaks: smaller at T = 86 °C (area 42.4 J g^−1^) and larger at T = 233.6 °C (area 369.2 J g^−1^), which indicate thermally cross-linked 2,3-dibro-2-propen-1-ol oligomer. At T = 242 °C, the oligomer underwent complete thermal degradation. 

#### 2.5.2. Reactions of Oligomerization Catalyzed by [VOO(dipic)(2-phepyH)] • H_2_O

The glass transition temperature values for the 3-buten-2-ol oligomer obtained using by [VOO(dipic)(2-phepyH)] • H_2_O as a precatalyst had the range ∆T = −86.7 to −75 °C (∆Cp = 0.325 J g^−1^ K^−1^). This step was followed by the transition of the oligomer from glass to an elastic state. During heating, two exothermic processes occurred. The first one took place at T = 114 °C (area 208 J g^−1^, height 0.556 mW mg^−1^) and at T = 222.3 °C (area 502 J g^−1^, height 4.312 mW mg^−1^), proving the deoligomerization reaction of 3-buten-2-ol.

Degradation of the allyl alcohol oligomer occurred in two steps, the first of which was at 151.9 °C (area 144 J g^−1^, height 0.401 mW mg^−1^). However, the second stage took place at 227.8 °C (area 490 J g^−1^, height 3.27 mW mg^−1^) and indicated the degradation of the allyl alcohol oligomer. Additionally, the transition of the 2-propen-1-ol oligomer from glass to elastic ∆T = −103 to −88 °C (∆Cp = 0.3 J g^−1^ K^−1^) can be observed in the DSC spectrum.

On the differential scanning calorimetry spectrum of 2,3-dibromo-2-propen-1-ol oligomer, three exothermic peaks can be observed: a smaller peak at 61.5 °C (with an area of 40.9 J g^−1^ and height 0.135 mW mg^−1^), and another at 230 °C (with an area of 322 J g^−1^, with a height of 2.81 mW mg^−1^). The last peak is at T = 448 °C (area 22.9 J g^−1^, height 0.09 mW mg^−1^), indicating combustion of the residual 2,3-dibromo-2-propen-1-ol oligomer.

#### 2.5.3. Reactions of Oligomerization Catalyzed by [VO(dipic)(dmbipy)] • 2 H_2_O

The decomposition of the test sample of the 3-buten-2-ol oligomer formed using [VO(dipic)(dmbipy)] • 2 H_2_O as a precatalyst was recorded as two large exothermic signals with peaks at T = 84 °C and approximately 156 °C.

The degradation of the allyl alcohol oligomer took place in one step at T = 118.5 °C and is probably indicative of deoligomerization of the allyl alcohol.

By analyzing the DSC spectrum of the 2,3-dibromo-2-propen-1-ol oligomer formed using [VO(dipic)(dmbipy)] • 2 H_2_O as a precatalyst, two exothermic peaks can be observed in which the oligomer decomposes, smaller at 135 °C and higher at ∆T = 82.5–107 °C.

#### 2.5.4. Reactions of Oligomerization Catalyzed by [VO(ODA)bipy] • 2 H_2_O

The decomposition of the test sample of the 3-buten-2-ol oligomer formed using [VO(dipic)(dmbipy)] • 2 H_2_O as a precatalyst was recorded as a large exothermic signal with an area of 469 J g^−1^, with a maximum at T = 222 °C. Other, weaker, exothermic signals were found at approximately T = 100 °C and 156 °C. By analyzing the DSC spectrum, it can be concluded that the 3-buten-2-ol oligomer changed from the elastic state to the glassy in the temperature range ∆T = −99 to −103 °C (∆Cp = 0.13 J g^−1^ K^−1^) [38,39,40].

The degradation of the allyl alcohol oligomer took place in two stages, the first of which was at ∆T = 145–161 °C. The second stage took place at ∆T = 214.6–236.3 °C and indicated the deoligomerization of the allyl alcohol oligomer. Additionally, the oligomer transition from elastic to glassy at ∆T = −91 to −79 °C (∆Cp = 0.25 J g^−1^ K^−1^) was distinguished.

The 2,3-dibromo-2-propen-1-ol oligomer changed from an elastic to glass state at ∆T = −89 to −74 °C (∆Cp = 0.21 J g^−1^ K^−1^). Additionally, we observed two exothermic peaks in which the oligomer decomposed, smaller at T = 110 °C (surface 132.4 J g^−1^) and larger at T = 218 °C (surface 240 J g^−1^). At T = 474 °C, the residue after combustion of the 2,3-dibromo-2-propen-1-ol oligomer underwent complete thermal decomposition.

### 2.6. Catalytic Activity of Green Precatalysts Based on Oxovanadium(IV) Coordination Compounds

In the final part of the research, the catalytic activities (Ca) of the oxovanadium(IV) complex compounds as precatalysts were determined using the formula (Table 7) [41,42,43]:(1)$$\mathrm{Ca}=\frac{\mathrm{mo}}{n_{\mathrm{VO}}\;\cdot t},$$
where mo is the weight of the oligomer sample [g], n_VO_ is the number of mmol of oxovanadium(IV) ions used in the oligomerization process [mmol] and t is the time of carrying out the oligomerization process [h].

We attempted to use oxovanadium(IV) microclusters with 2-phenylpyridine (oxovanadium(IV) cage) as precatalysts with AlEt_2_Cl and MAO (methylaluminoxane), but these catalytic systems were inactive. We did not observe any changes in the ongoing reaction. All green precatalysts (compounds 1–5) were inactive in the oligomerization reaction of 2-chloro-2-propen-1-ol, with AlEt_2_Cl as the cocatalyst.

## 3. Materials and Methods

### 3.1. Materials

All chemical compounds: vanadyl acetylacetonate, vanadyl sulfate pentahydrate, 1,10-phenanthroline, thiodiacetate, oxydiacetic acid, 2-phenylpyridine, dipicolinic acid, 4,4′-dimethoxy-2,2′-bipyridine, 2,2′-bipyridine, n-hexane, ethylaluminium dichloride, dimethyl sulfoxide, 3-buten-2-ol, allyl alcohol, 2-chloro-2-propen-1-ol and 2,3-dibromo-2-propen-1-ol were purchased from Merck, Darmstad, Germany. The purity of the reagents was in the range of 97–100%.

### 3.2. Elemental Analysis

Elemental analysis (the Vario El Cube apparatus, Langenselbold, Germany) was conducted on samples (2 mg) of the complexes that were homogeneous and dry.

### 3.3. FTIR Spectra

The IR spectra were recorded in the range 4000–400 cm^−1^ on a KBr pastil. DLATGS (Branch Überlingen, Germany) was used as a detector. The IFS66 apparatus by BRUKER (Branch Überlingen, Germany) had a resolution of 0.12 cm^−1^.

### 3.4. MALDI-TOF-MS Spectra

MALDI-TOF-MS spectra were recorded on the Bruker Biflex III company (Branch Überlingen, Germany). 2,5-Dihydroxybenzoic acid (DHB) and α-cyano-4-hydroxycinnamic acid (CCA) were used as a matrix.

### 3.5. Thermogravimetric Analysis (TGA)

Thermogravimetric (TG) analysis was performed on the NETZSCH TG 209 instrument in the temperature range from room temperature to 600 °C in an oxidation atmosphere. The mass of samples subjected to thermal analysis was approximately 5 mg. The morphologies of the as-prepared samples were investigated using field-emission SEM using JSM-7610F, JEOL.

### 3.6. The Differential Scanning Calorimetry (DSC) 

Differential scanning calorimetry (DSC) studies were performed using equipment from Mettler Toledo in accordance with the following program: cooling the isothermal segment to −150 °C, keeping the segment at a constant temperature of −150 °C for a period of 5 min and finally, heating the segment until the temperature facilitates the decomposition of the sample. The analysis was performed in an atmosphere of nitrogen. The sample for the DSC measurements was approximately 5 mg. Calibration was carried out based on standards for thermal analysis (ind, n-octane). The liquid nitrogen was used to cool.

### 3.7. Synthesis of Oxovanadium(IV) Coordination Complex Compounds

#### 3.7.1. Synthesis of [VO(TDA)phen] • 1.5 H_2_O

The synthesis of [VO(TDA)(phen)] • 1.5 H_2_O was carried out according to the procedures described in the literature [26,27]. The results of elemental analysis are as follows: 44.47% C, 3.55% H, 6.49% N and 7.27% S; analysis calculations included 43.64% C, 3.41% H, 6.36% N and 7.27% S.

#### 3.7.2. Synthesis of Oxovanadium(IV) Cage (Oxovanadium(IV) Microclusters with 2-phenylpyridine)

The synthesis of oxovanadium(IV) microclusters with 2-phenylpyridine was carried out according to the procedures described in the literature [28]. 

#### 3.7.3. Synthesis of [VOO(dipic)(2-phepyH)] • H_2_O

The synthesis of [VOO(dipic)(2-phepyH)] • H_2_O was carried out according to the procedures described in the literature [29]. 

#### 3.7.4. Synthesis of [VO(dipic)(dmbipy)] • 2 H_2_O

The synthesis of [VO(dipic)(dmbipy)] • 2 H_2_O was carried out according to the procedures described in the literature [29]. The results of elemental analysis are as follows: 47.46% C, 3.69% H and 8.77% N; analysis calculations included 47.10% C, 3.93% H and 8.68% N.

#### 3.7.5. Synthesis of [VO(ODA)bipy] • 2 H_2_O

The synthesis of [VO(ODA)bipy] • 2 H_2_O was carried out according to the procedures described in the literature [30,31]. The results of elemental analysis are as follows: 42.04% C and 3.32% H; analysis calculations included 42.94% C and 4.09% H.

### 3.8. Oligomerization Process

First, 3 micromoles of a complex compound based on oxovanadium(IV) were dissolved in 1 mL of DMSO in an ultrasonic bath. Then, 1 mL of n-hexane was poured into the vial and sealed with a rubber stopper against air. The reaction mixture was then saturated with nitrogen for 15 min, and the activator was charged with 3 mL of ethylaluminium dichloride (AlEtCl_2_) solution and 3 mL of monomer. Finally, the oligomers were washed with a mixture of 1 M hydrochloric acid and 1 M methanol.

## 4. Conclusions

The catalytic activities of the vanadium(IV, V) complex compounds we studied are summarized in Figure 2, Figure 3 and Figure 4. During the oligomerization of 3-buten-2-ol, the highest catalytic activity was shown by the dioxovanadium(V) complex compound with 2-phenylpyridine and dipicolinate anion (Figure 2). The dipicolinate oxovanadium(IV) complex compound with 4,4’-dimethoxy-2,2’-bipyridyl showed similar catalytic activity to the thiodiacetate oxovanadium(IV) complex compound with 1,10-phenanthroline in the oligomerization of 3-buten-2-ol. However, comparing the obtained values of catalytic activities with the available literature data on the activity of vanadium(IV, V) compounds in the oligomerization of 3-buten-2-ol, the oxovanadium(IV) microclusters with 2-phenylpyridine showed higher catalytic activity than the complex compound dioxovanadium(V) with 2-phenylpyridine and dipicolinate anion. This may suggest that microclusters act more efficiently than small-molecule vanadium(IV,V) complexes in the oligomerization of 3-buten-2-ol.

The dioxovanadium(V) complex compound with 2-phenylpyridine and dipicolinate anion proved to be the best precatalyst in the oligomerization of 2-propen-1-ol as was also the case in the oligomerization of 3-buten-2-ol (Figure 3). The analysis of literature data on the oligomerization of 2-propen-1-ol showed that, in the case of vanadium(IV,V) complex compounds, the presence of so-called additional ligands in addition to the polycarboxylate anion significantly improves the catalytic activity of a given complex compound. Moreover, it was also be seen that the choice of activator plays a significant role in the catalytic performance of the complex compound. The use of MMAO-12 as an activator reduces the catalytic activity of the complex compound by 15 times compared to the use of ethylaluminum dichloride (AlEtCl_2_) as an activator.

The use of vanadium(IV,V) complexes for 2,3-dibromo-2-propen-1-ol oligomerization (Figure 4) provides similar results as in the oligomerization of 3-buten-2-ol. The highest catalytic activity was possessed by the dioxovanadium(V) complex compound with 2-phenylpyridine and dipicolinate anions. On the other hand, the lowest catalytic activity in the oligomerization of 2,3-dibromo-2-propen-1-ol was presented by the oxovanadium(IV) complex compound with the thiodiacetate anion and 1,10-phenanthroline. Based on the study, it can be concluded that the high degree of oxidation of vanadium in the complex, i.e., the V oxidation degree, promotes the high value of catalytic activity of the complex compound in different types of olefin oligomerizations: 2,3-dibromo-2-propen-1-ol, 2-propen-1-ol and 3-buten-2-ol oligomerizations. 

In the literature other oxovanadium(IV) complexes chelated with Schiff’s base such as salen (H_2_salen = N,N′-ethylenebis(salicylideneimine)), acacen (H_2_acacen=N,N′-ethylenebis(acetylacetonimine)), aceten (H_2_aceten = N,N′-ethylenebis(2-hydroxyacetophenoneimine)) and acetph (H_2_acetph = N,N′-phenylene-1,2-bis(2-hydroxyacetophenoneimine)) are known. Precatalysts are characterized by moderate catalytic activity in the polymerization reaction of ethylene after activation by EtAlCl_2_. Their activity increased as follows: [VO(salen)] < [VO(acetph)] < [VO(aceten)] < [VO(acacen)] [45].

## Figures and Tables

**Figure 1 molecules-27-08038-f001:**
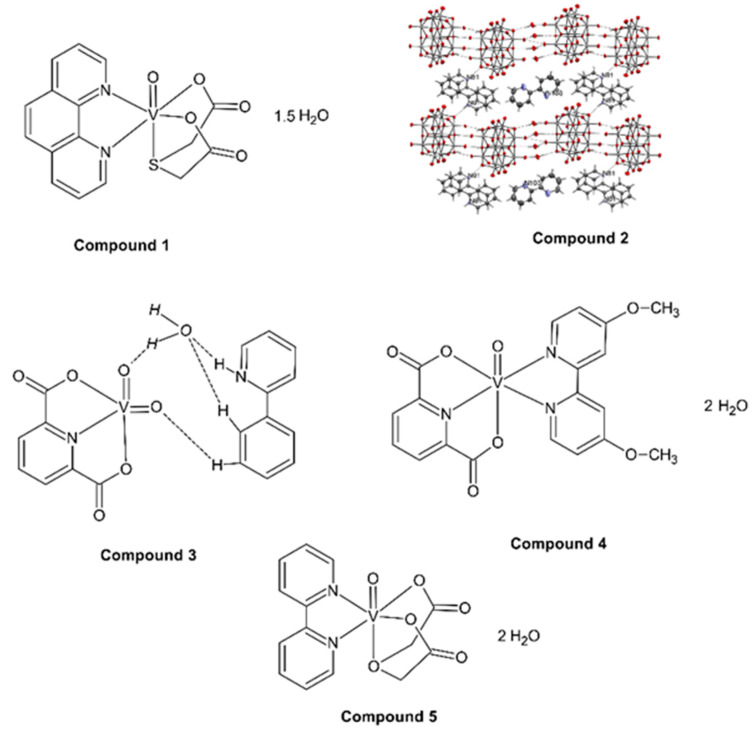
Molecular formulas of [VO(TDA)phen] • 1.5 H_2_O (Compound 1) (TDA = thiodiacetate), (phen = 1,10-phenanthroline), oxovanadium(IV) microclusters with 2-phenylpyridine (Compound 2), [VOO(dipic)(2-phepyH)] • H_2_O (Compound 3) (dipic = pyridine-2,6-dicarboxylate anion), (2-phepyH = 2-phenylpyridine), [VO(dipic)(dmbipy)] • 2H_2_O (Compound 4), (dmbipy = 4,4′-dimethoxy-2,2′-dipyridyl) and [VO(ODA)(bipy)] • 2 H_2_O (Compound 5) (ODA = oxydiacetate), (bipy = 2,2′-bipyridine).

**Figure 2 molecules-27-08038-f002:**
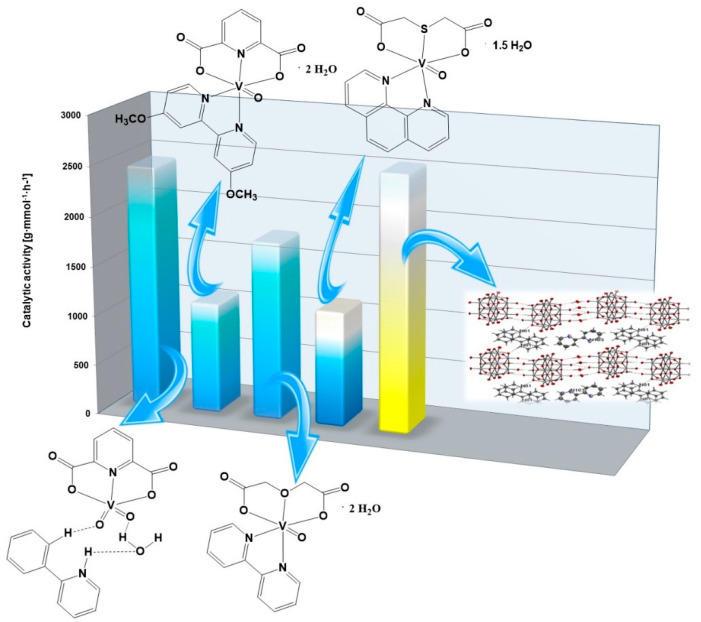
The catalytic activities of vanadium(IV,V) complexes for 3-buten-2-ol oligomerization (based on this work and [28]).

**Figure 3 molecules-27-08038-f003:**
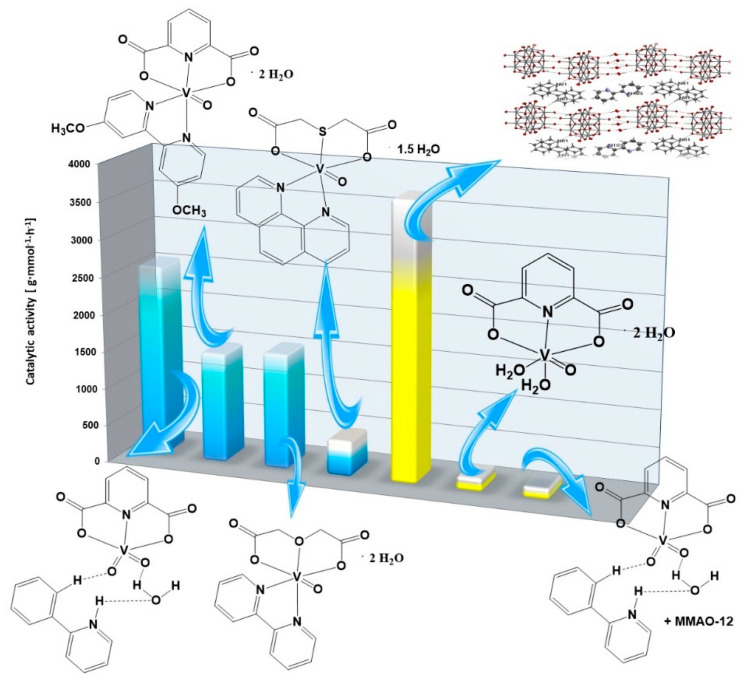
The catalytic activities of vanadium(IV,V) complexes for 2-propen-1-ol oligomerization (based on this work and [28,35,43,44]).

**Figure 4 molecules-27-08038-f004:**
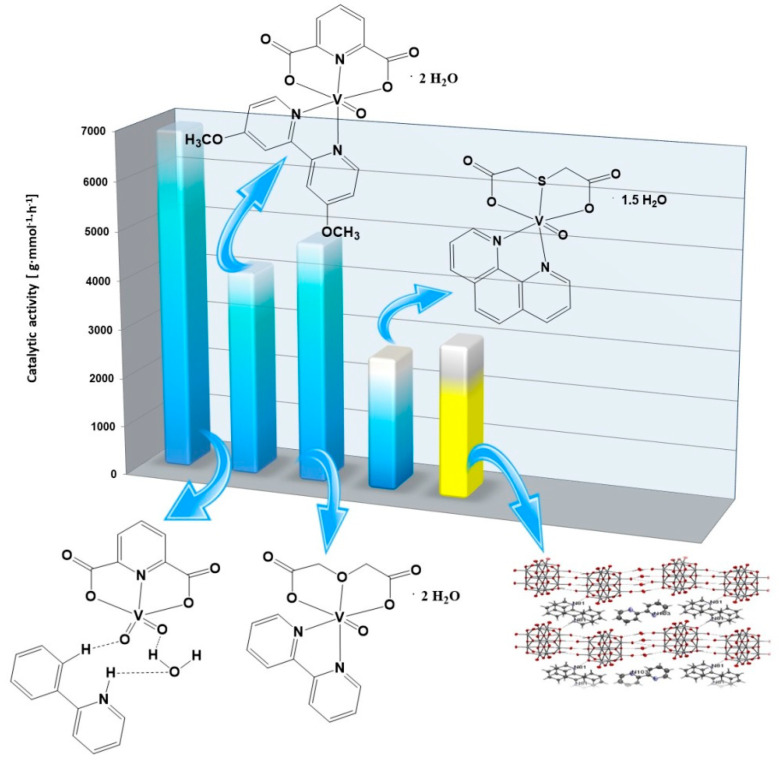
The catalytic activities of vanadium(IV,V) complexes for 2,3-dibromo-2-propen-1-ol oligomerization (based on this work and [28]).

**Table 1 molecules-27-08038-t001:** The results of FTIR spectroscopy for the products of 3-buten-2-ol oligomerization.

Complex Compound	Wavenumber [cm^−1^]	Type of Vibration with a Function Group
[VO(TDA)phen] • 1.5 H_2_O	3454	Strong -OH stretching vibrations
	2943	Weak asymmetric -CH stretching vibrations
	1389	Very weak -CH_2_ bending vibrations
	1074	Strong C-O stretching vibrations
	764	Very weak -CH_2_ rocking vibrations
[VOO(dipic)(2-phepyH)] • H_2_O	3488	Strong symmetric -OH stretching vibrations
	2988	Strong asymmetric -CH stretching vibrations
	1458	Strong -CH_2_ bending vibrations
	1088	Strong C-O stretching vibrations
	746	Weak -CH_2_ rocking vibrations
[VO(dipic)(dmbipy)] • 2 H_2_O	3468	Strong symmetric -OH stretching vibrations
	2966	Strong asymmetric -CH stretching vibrations
	1492	Strong -CH_2_ bending vibrations
	1098	Strong C-O stretching vibrations
	733	Weak -CH_2_ rocking vibrations
[VO(ODA)bipy] • 2 H_2_O	3399	Symmetric -OH stretching vibrations
	2998	Strong asymmetric -CH stretching vibrations
	1488	Weak -CH_2_ bending vibrations
	1092	Strong C-O stretching vibrations
	734	Very weak -CH_2_ rocking vibrations

**Table 2 molecules-27-08038-t002:** The results of FTIR spectroscopy for the products of allyl alcohol oligomerization.

Complex Compound	Wavenumber [cm^−1^]	Type of Vibration with Function Group
[VO(TDA)phen] • 1.5 H_2_O	3488	Strong symmetric -OH stretching vibrations
	2988	Strong asymmetric -CH stretching vibrations
	1420	-CH_2_ bending vibrations
	1101	Strong C-O stretching vibrations
	724	Very weak -CH_2_ rocking vibrations
[VOO(dipic)(2-phepyH)] • H_2_O	2998	Strong asymmetric -CH stretching vibrations
	1436	Strong -CH_2_ bending vibrations
	1079	Strong C-O stretching vibrations
	746	Weak -CH_2_ rocking vibrations
[VO(dipic)(dmbipy)] • 2 H_2_O	3449	Strong symmetric -OH stretching vibrations
	2985	Strong asymmetric -CH stretching vibrations
	1435	Strong -CH_2_ bending vibrations
	1104	Strong C-O stretching vibrations
	721	Weak -CH_2_ rocking vibrations
[VO(ODA)bipy] • 2 H_2_O	3217	Weak -OH stretching vibrations
	2998	Asymmetric -CH stretching vibrations
	1477	Weak -CH_2_ bending vibrations
	1091	Strong C-O stretching vibrations
	718	Very weak -CH_2_ rocking vibrations

**Table 3 molecules-27-08038-t003:** The results of FTIR spectroscopy for the products of 2,3-dibromo-2-propen-1-ol oligomerization.

Complex Compound	Wavenumber [cm^−1^]	Type of Vibration with a Function Group
[VO(TDA)phen] • 1.5 H_2_O	3482	Symmetric -OH stretching vibrations
	2945	Strong asymmetric -CH stretching vibrations
	1419	-CH_2_ bending vibrations
	1093	Strong C-O stretching vibrations
	731	Very weak -CH_2_ rocking vibrations
[VOO(dipic)(2-phepyH)] • H_2_O	3432	Strong symmetric -OH stretching vibrations
	2935	Strong asymmetric -CH stretching vibrations
	1476	Strong -CH_2_ bending vibrations
	1096	Strong C-O stretching vibrations
	755	Very weak -CH_2_ rocking vibrations
[VO(dipic)(dmbipy)] • 2 H_2_O	3566	Symmetric -OH stretching vibrations
	2921	Asymmetric -CH stretching vibrations
	1411	Very weak -CH_2_ bending vibrations
	1081	Strong C-O stretching vibrations
	762	Very weak -CH_2_ rocking vibrations
[VO(ODA)bipy] • 2 H_2_O	3279	Symmetric -OH stretching vibrations
	2963	Strong asymmetric -CH stretching vibrations
	1421	-CH_2_ bending vibrations
	1093	Strong C-O stretching vibrations
	713	Very weak -CH_2_ rocking vibrations

**Table 4 molecules-27-08038-t004:** Degree of oligomerization for 3-buten-2-ol oligomers obtained using oxovanadium(IV) precatalysts.

Complex Compound	Degree of Oligomerization (n)
[VO(TDA)phen] • 1.5 H_2_O	5, 9, 11, 14
[VOO(dipic)(2-phepyH)] • H_2_O	3, 6
[VO(dipic)(dmbipy)] • 2 H_2_O	3, 5, 6
[VO(ODA)bipy] • 2 H_2_O	5, 6, 9, 12, 16

**Table 5 molecules-27-08038-t005:** Degree of oligomerization for 2-propen-1-ol oligomers obtained using oxovanadium(IV) precatalysts.

Complex Compound	Degree of Oligomerization (n)
[VO(TDA)phen] • 1.5 H_2_O	7, 11, 14, 17, 20
[VOO(dipic)(2-phepyH)] • H_2_O	7, 11, 14, 17
[VO(dipic)(dmbipy)] • 2 H_2_O	2, 4, 6, 7, 8, 9, 12
[VO(ODA)bipy] • 2 H_2_O	7, 9, 10, 11, 14, 17, 20

**Table 6 molecules-27-08038-t006:** Degree of oligomerization for 2,3-dibromo-2-propen-1-ol oligomers obtained using oxovanadium(IV) precatalysts.

Complex Compound	Degree of Oligomerization (n)
[VO(TDA)phen] • 1.5 H_2_O	2, 3, 4
[VOO(dipic)(2-phepyH)] • H_2_O	2, 3, 4
[VO(dipic)(dmbipy)] • 2 H_2_O	2, 3, 4
[VO(ODA)bipy] • 2 H_2_O	2, 3, 4, 5

**Table 7 molecules-27-08038-t007:** Influence of coordination complexes based on oxovanadium(IV) on catalytic activity in oligomerization reactions of allyl alcohol, 3-buten-2-ol and 2,3-dibromo-2-propen-1-ol.

Complex Compounds	Olefins	Activator	Amount of Oxovanadium(IV) [mmol]	The Molar Ratio of the Complex Compound to the Activator	Temperature [°C]	Catalytic Activity [g mmol^−1^ h^−1^]
[VO(TDA)phen] • 1.5 H_2_O	Allyl alcohol	AlEtCl_2_	0.003	1:1000	30	492
	3-buten-2-ol					1160
	2,3-dibromo-2-propen-1-ol					2707
[VOO(dipic)(2-phepyH)] • H_2_O	Allyl alcohol					2591
	3-buten-2-ol					2441
	2,3-dibromo-2-propen-1-ol					6933
[VO(dipic)(dmbipy)] • 2H_2_O	Allyl alcohol					1494
	3-buten-2-ol					1121
	2,3-dibromo-2-propen-1-ol					4188
[VO(ODA)(bipy)] • 2 H_2_O	Allyl alcohol					1572
	3-buten-2-ol					1792
	2,3-dibromo-2-propen-1-ol					4894

## Data Availability

On request of those interested.

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
