# Peer review of "A Series of Green Oxovanadium(IV) Precatalysts with O, N and S Donor Ligands in a Sustainable Olefins Oligomerization Process"

_molecules, 2022, doi:10.3390/molecules27228038_

Round 1

Reviewer 1 Report

The authors used reported vanadium catalysts for the polymerization of various polar olefins. The authors claim that these catalysts are highly active in the oligomerization of these olefins. Here the authors need to give the tables for the specific  yields and reaction conditions. In general vanadium catalysts, similar to Ziegler-Natta catalysts, are very sensitive to polar groups and the ability to oligomerize these polar olefins here is needed a reasonable explanation from the author.Another more serious issue is that the authors do not give the structure and degree of polymerization of these oligomers. The authors are recommended to provide detailed NMR analysis of these oligomers. On the whole this work lacks some detailed data and characterization to illustrate the points claimed in the paper.

Author Response

Answers to the Reviewers’ comments

We are very grateful to the Reviewers for their time and constructive comments on our manuscript. We have implemented their comments and suggestions and wish to submit a revised version of the manuscript for further consideration in the Molecules. Changes in the initial version of the manuscript are highlighted in yellow (Reviewer 1) in the revised version. Below, we also provide a point-by-point response explaining how we have addressed each of the Reviewers’ comments.

Answers to the Reviewer #1:

Comment:

     The authors used reported vanadium catalysts for the polymerization of various polar olefins. The authors claim that these catalysts are highly active in the oligomerization of these olefins. Here the authors need to give the tables for the specific  yields and reaction conditions. In general vanadium catalysts, similar to Ziegler-Natta catalysts, are very sensitive to polar groups and the ability to oligomerize these polar olefins here is needed a reasonable explanation from the author. Another more serious issue is that the authors do not give the structure and degree of polymerization of these oligomers. The authors are recommended to provide detailed NMR analysis of these oligomers. On the whole this work lacks some detailed data and characterization to illustrate the points claimed in the paper.

Authors' Response:

     Table 7 summarizing the results of the research for reaction conditions is presented on page 11. The oligomerization reaction was carried out at a temperature of 30 ℃, in a nitrogen atmosphere and at ambient pressure. In each case, 3 micromoles of precatalyst were used. The molar ratio between precatalyst and activator (AlEtCl2) V:Al was 1:1000.

Complex Compounds

Olefins

Activator

Amount of oxovanadium(IV) [mmol]

The molar ratio of the complex compound to the activator

Temperature [℃]

Catalytic activity [g · mmol−1 · h−1]

[VO(TDA)phen] • 1.5 H2O

Allyl alcohol

AlEtCl2

0.003

1:1000

30

492.26

3-buten-2-ol

1160.1

2,3-dibromo-2-propen-1-ol

2707.13

[VOO(dipic)(2-phepyH)] • H2O

Allyl alcohol

2591.47

3-buten-2-ol

2440.67

2,3-dibromo-2-propen-1-ol

6932.53

[VO(dipic)(dmbipy)] • 2H2O

Allyl alcohol

1494.4

3-buten-2-ol

1120.53 

2,3-dibromo-2-propen-1-ol

4187.73

[VO(ODA)(bipy)] • 2 H2O

Allyl alcohol

1572.09 

3-buten-2-ol

1792.32

2,3-dibromo-2-propen-1-ol

4894.4

     The degree of oligomerization was tested by performing MALDI-TOF-MS tests. Using this technique, we are able to determine in a very precise way how many units the oligomer mixture is made of, i.e. what is its degree of oligomerization. For example, the degree of oligomerization of 3-buten-2-ol oligomer obtained by using [VO(TDA)phen] • 1.5 H2O is n=5, 9, 11, 14. We have prepared tables (Table 4-6) describing the degree of oligomerization (n).

Table 4. Degree of oligomerization for 3-buten-2-ol oligomers obtained using oxovanadium(IV) precatalysts.

Complex Compound

Degree of oligomerization (n)

[VO(TDA)phen] • 1.5 H2O    

5, 9, 11, 14

[VOO(dipic)(2-phepyH)] • H2O

3, 6

[VO(dipic)(dmbipy)] • 2 H2O

3, 5, 6

[VO(ODA)bipy] • 2 H2O

5, 6, 9, 12, 16

Table 5. Degree of oligomerization for 2-propen-1-ol oligomers obtained using oxovanadium(IV) precatalysts.

Complex Compound

Degree of oligomerization (n)

[VO(TDA)phen] • 1.5 H2O    

7, 11, 14, 17, 20

[VOO(dipic)(2-phepyH)] • H2O

7, 11, 14, 17

[VO(dipic)(dmbipy)] • 2 H2O

2, 4, 6, 7, 8, 9, 12

[VO(ODA)bipy] • 2 H2O

7, 9, 10, 11, 14, 17, 20

Table 6. Degree of oligomerization for 2,3-dibromo-2-propen-1-ol oligomers obtained using oxovanadium(IV) precatalysts.

Complex Compound

Degree of oligomerization (n)

[VO(TDA)phen] • 1.5 H2O    

2, 3, 4

[VOO(dipic)(2-phepyH)] • H2O

2, 3, 4

[VO(dipic)(dmbipy)] • 2 H2O

2, 3, 4

[VO(ODA)bipy] • 2 H2O

2, 3, 4, 5

        We agree that the use of polar monomers can be problematic in obtaining a high molecular weight polymer due to the suppressed propagation step. This may also be a determining factor in obtaining an oligomer rather than a polymer. The reason may be the coordination of the polar groups to the metal ion of the precatalyst, creating a chelate structure or the formation of stable π-allyl species (vide infra), which does not allow for the next stage of chain growth. However, the study of the mechanism of the oligomerization reaction was not the subject of our study.

There are many scientific papers that confirm that the use of polar monomers in the olefin polymerization reaction is possible.

Takeuchi, D. (2013). Transition metal-catalyzed polymerization of polar allyl and diallyl monomers. MRS Bulletin, 38(03), 252–259. doi:10.1557/mrs.2013.51 

Chesworth, A. G., Haszeldine, R. N., & Tait, P. J. T. (1974). Polymerization of vinyl chloride by use of modified Ziegler–Natta catalysts. I. Overall kinetic features. Journal of Polymer Science: Polymer Chemistry Edition, 12(8), 1703-1716.

Huang, J., & Rempel, G. L. (1995). Ziegler-Natta catalysts for olefin polymerization: mechanistic insights from metallocene systems. Progress in Polymer Science, 20(3), 459-526.

     The aim of our research was to investigate the catalytic activity of green complex compounds based on oxovanadium(IV). Performing additional NMR analyses in such a short time set by the journal is impossible. The characteristic groupings of the oligomers were examined by the FTIR technique.

Reviewer 2 Report

The submission reports a series of vanadium complexes as catalysts for polymerization, and results are somewhat interesting to readers.  It is recommended with some revisions with considering followings:

1  The description of polyolefin industry is not necessary with some errors in the amounts, probably authors should double check and revise (remove) them. 

2  Though there are some scientists considering vanadium catalysts similar to titanium catalysts named as Ziegler-Natta catalyst, it is not necessary to emphasize such issue (key word). 

3 There are complex catalysts of vanadium reviewed recently, Coordination Chemistry Reviews, 2020, 416, 213332, I am not sure whether authors crossed it or not. 

4  In my understanding, the vanadium complexes with water molecules were assumed on the basis of elemental analysis. Have authors tried to dry them or not? In addition, they are not good points for catalytic performance because of consuming some amount of aluminium reagents.

5  The resultant polymers were measured with digital reports, however, authors should keep three efficient digits instead of reported ones. 

In general, it is recommended for its acceptance, but it would be also better to optimize the catalytic conditions because it is common for polymerization with various conditions having an optimal condition.    

Author Response

We are very grateful to the Reviewers for their time and constructive comments on our manuscript. We have implemented their comments and suggestions and wish to submit a revised version of the manuscript for further consideration in the Molecules. Changes in the initial version of the manuscript are highlighted in green (Reviewer 2) in the revised version. Below, we also provide a point-by-point response explaining how we have addressed each of the Reviewers’ comments.

Answers to the Reviewer #2:

Comment:

     The submission reports a series of vanadium complexes as catalysts for polymerization, and results are somewhat interesting to readers.

Authors' Response:

    Thank you very much.

Comment:

      The description of polyolefin industry is not necessary with some errors in the amounts, probably authors should double check and revise (remove) them.

Authors' Response:

     With reference to the reviewer's comment, the amounts have been removed.

Comment:

      Though there are some scientists considering vanadium catalysts similar to titanium catalysts named as Ziegler-Natta catalyst, it is not necessary to emphasize such issue (key word).

Authors' Response:

    According to the reviewer's comment, this keyword has been removed.

Comment:

      There are complex catalysts of vanadium reviewed recently, Coordination Chemistry Reviews, 2020, 416, 213332, I am not sure whether authors crossed it or not.

Authors' Response:

   Thank you very much for recommending the article. The review paper influenced the quality of the discussed results in the conclusion. We have added the following paragraph in the revised version of the manuscript:

In the literature other oxovanadium(IV) complexes chelated with Schiff's base such as salen (H2salen = N,N′-ethylenebis(salicylideneimine)), acacen (H2acacen=N,N′-ethylenebis(acetylacetonimine)), aceten (H2aceten = N,N′-ethylenebis(2-hydroxyacetophenoneimine)), acetph (H2acetph = N,N′-phenylene-1,2-bis(2-hydroxyacetophenoneimine)) are known. Precatalysts are characterized by moderate catalytic activity in the polymerization reaction of ethylene after activation by EtAlCl2. Their activity increased as follows: [VO(salen)] < [VO(acetph)] < [VO(aceten)] < [VO(acacen)] [45, 46].

[45] Phillips, A. M. F.; Suo, H.; Maria de Fátima, C.; Pombeiro, A. J.; Sun, W. H. Recent developments in vanadium-catalyzed olefin coordination polymerization. Coord. Chem. Rev. 2020, 416, 213332. https://doi.org/10.1016/j.ccr.2020.213332.

[46] BiaÅ‚ek, M.; Leksza, A.; Piechota, A.; Kurzak, K.; Koprek, K. Oxovanadium(IV) complexes with [ONNO]-chelating ligands as catalysts for ethylene homo-and copolymerization. J. Polym. Res. 2014, 21, 1. https://doi.org/10.1007/s10965-014-0389-4.

Comment:

       In my understanding, the vanadium complexes with water molecules were assumed on the basis of elemental analysis. Have authors tried to dry them or not? In addition, they are not good points for catalytic performance because of consuming some amount of aluminium reagents.

Authors' Response:

    Oxovanadium(IV) complex compounds were dried and in the form of powders were used as precatalysts in the olefin oligomerization reaction. Finally, the oligomers were washed with a mixture of 1 M hydrochloric acid and 1 M methanol to purify and get rid of residual precatalysts.

Comment:

  The resultant polymers were measured with digital reports, however, authors should keep three efficient digits instead of reported ones.

Authors' Response:

    According to the reviewer's suggestions, all numerical values have been reduced to three efficient digits.

Comment:

     In general, it is recommended for its acceptance, but it would be also better to optimize the catalytic conditions because it is common for polymerization with various conditions having an optimal

Authors' Response:

    Thank you very much. It is not possible to optimize the process on such a small scale because the oligomerization process is carried out in a 10 ml vial.

-1 mL determines the solvent used in the activator, in the case of AlEtCl2 it is hexane

-1 mL the best solvent in which to dissolve the complex compound, in this case DMSO (because they did not dissolve in hexane).

-3 mL of activator

-3 mL monomer

Therefore, the only optimization option is the temperature effect. However, we wanted to use the mildest possible reaction conditions, hence the reaction was carried out at 30 ℃ and no reduced pressure.

Round 2

Reviewer 1 Report

Although the authors were unable to perform NMR characterization of the resulting oligomers due to objective reasons, the information provided by NMR is still necessary. In addition the authors needed to give the approximate structure of the resulting oligomers. If the editor feels that the experiment does not need to be added, this manuscript can be recommended for publication.